# Magnetic Resonance Planimetry in the Differential Diagnosis between Parkinson’s Disease and Progressive Supranuclear Palsy

**DOI:** 10.3390/brainsci12070949

**Published:** 2022-07-20

**Authors:** Andrea Quattrone, Maurizio Morelli, Maria G. Bianco, Jolanda Buonocore, Alessia Sarica, Maria Eugenia Caligiuri, Federica Aracri, Camilla Calomino, Marida De Maria, Maria Grazia Vaccaro, Vera Gramigna, Antonio Augimeri, Basilio Vescio, Aldo Quattrone

**Affiliations:** 1Institute of Neurology, University “Magna Graecia”, 88100 Catanzaro, Italy; an.quattrone@hotmail.it (A.Q.); m.morelli@unicz.it (M.M.); jolandabuonocore@gmail.com (J.B.); 2Department of Medical and Surgical Sciences, University “Magna Graecia”, 88100 Catanzaro, Italy; maria.giovanna.bianco@gmail.com; 3Neuroscience Research Center, University “Magna Graecia”, 88100 Catanzaro, Italy; alessia.sarica@gmail.com (A.S.); me.caligiuri@unicz.it (M.E.C.); fedearacri@gmail.com (F.A.); camillacalomino@gmail.com (C.C.); marida.demaria@gmail.com (M.D.M.); mg.vaccaro@unicz.it (M.G.V.); veragramigna@gmail.com (V.G.); 4Biotecnomed S.C.aR.L., 88100 Catanzaro, Italy; antonio.augimeri@biotecnomed.it; 5Institute of Molecular Bioimaging and Physiology, National Research Council (IBFM-CNR), 88100 Catanzaro, Italy; basilio.vescio@ibfm.cnr.it

**Keywords:** Parkinson’s disease, progressive supranuclear palsy, MR planimetry, MRPI, biomarkers

## Abstract

The clinical differential diagnosis between Parkinson’s disease (PD) and progressive supranuclear palsy (PSP) is often challenging. The description of milder PSP phenotypes strongly resembling PD, such as PSP-Parkinsonism, further increased the diagnostic challenge and the need for reliable neuroimaging biomarkers to enhance the diagnostic certainty. This review aims to summarize the contribution of a relatively simple and widely available imaging technique such as MR planimetry in the differential diagnosis between PD and PSP, focusing on the recent advancements in this field. The development of accurate MR planimetric biomarkers, together with the implementation of automated algorithms, led to robust and objective measures for the differential diagnosis of PSP and PD at the individual level. Evidence from longitudinal studies also suggests a role of MR planimetry in predicting the development of the PSP clinical signs, allowing to identify PSP patients before they meet diagnostic criteria when their clinical phenotype can be indistinguishable from PD. Finally, promising evidence exists on the possible association between MR planimetric measures and the underlying pathology, with important implications for trials with new disease-modifying target therapies.

## 1. Introduction

Differentiating Parkinson’s disease (PD) from progressive supranuclear palsy (PSP) can be challenging in the early stage of the disease when the symptoms are few and mild and the response to dopaminergic treatment is still uncertain [1,2,3,4]. This is especially true in PD patients presenting with prevalent gait disorder and balance disturbances rather than tremor (postural instability and gait disorder, PIGD phenotype) [5,6]. The vertical supranuclear gaze palsy (VSGP) is the clinical sign with the highest specificity for PSP pathology [7,8,9], and it is often the clinical sign leading the differential diagnosis between PSP and other parkinsonian syndromes [10,11]. However, large pathological studies demonstrated that VSGP can appear extremely late during the course of the disease, up to 19 years after the PSP clinical onset [12], making the early diagnosis often difficult. The VSGP is often preceded by a reduction of amplitude and velocity of vertical saccades [8,11,13,14], but the presence of mild ocular slowness can be extremely challenging to evaluate also for movement disorders specialists due to the interindividual variability, the eyelids limiting the assessment of downward saccades and the up-gaze restriction, which can be at times observed in PD patients and elders [15,16]. The differential diagnosis between PD and PSP is even more complex because the PSP clinical spectrum is quite broad, including not only the classic PSP phenotype with early oculomotor abnormalities and falls, termed PSP-Richardson’s syndrome (PSP-RS), but also several other PSP subtypes which often show lower clinical severity and progression rate than PSP-RS [9,11,17]. For example, PSP-P, which is the most frequent phenotype after PSP-RS, has a clinical phenotype mainly characterized by parkinsonism, making the differential diagnosis with PD and the parkinsonian variant of multiple system atrophy extremely challenging [11,17,18,19,20].

To date, several MR imaging biomarkers have proven to be useful in distinguishing between PD and PSP and can be used to support the clinical diagnosis [17,21,22]. Most of them rely on structural neuroimaging, aiming to evaluate the atrophy of brainstem structures involved in PSP and spared in PD [17,21,22]. In this review, we discuss the most important MR planimetric quantitative biomarkers developed so far, distinguishing between simple manual measurements and automated objective biomarkers. We also focus on recent advancements in the field, including the role of MR planimetry in the early stage of the disease and the association between imaging findings and PSP pathology.

## 2. MR Planimetric Biomarkers

Structural brain imaging is usually normal in PD patients, but it has a crucial role in ruling out other conditions which can be misdiagnosed as PD, such as atypical or secondary parkinsonisms [23,24]. The brain MR imaging feature typical of PSP is the atrophy of the midbrain and superior cerebellar peduncles (SCPs), coupled with the enlargement of the third ventricle [24,25,26]. The midbrain tegmentum atrophy can lead to the well-known ‘hummingbird sign’ (flat or concave midbrain tegmentum with preserved pontine volume forming the silhouette of a hummingbird or king penguin on sagittal MR images) [21,25,27] and ‘morning glory sign’ (concavity of the lateral margin of the midbrain tegmentum resembling a lateral view of the morning glory flower on axial MR images) [21,27,28], which are the two main radiological signs of PSP (Figure 1). The qualitative evaluation of MR images, however, lacks objectivity and has suboptimal sensitivity (50–68%) for PSP diagnosis, with a large percentage of patients not showing these imaging signs at the beginning of the disease [26,27].

### 2.1. Simple Manual Linear Measurements

In the last two decades, some simple linear measurements that can be performed directly on MR images by neurologists or neuroradiologists have demonstrated good performances in distinguishing PSP from PD. These measurements are typically performed on T1-weighted images, which are part of the routine MR protocol and provide excellent grey and white matter contrast, allowing to evaluate anatomical details [23,24]. The antero-posterior (AP) midbrain diameter was originally proposed around 20 years ago to differentiate PSP from PD and multiple system atrophy (MSA) [29,30], and its usefulness has been confirmed in a few other studies [31,32,33,34]. To correct for the inter-individual variability, the AP midbrain diameter can be normalized by the AP pontine diameter (midbrain-to-pons [M/P] diameter ratio). A couple of studies [35,36] demonstrated that the M/P diameter ratio (M/P < 0.52 in the original study [35]) had excellent (100%) specificity but suboptimal (66.7%) sensitivity in distinguishing PSP from PD, MSA, and controls in two small cohorts of patients with pathologically proven diagnoses. The diagnostic performances of these measures, however, have been tested only in a few single-center studies with variable results, probably due to the small sample size and different measurement techniques (on axial or sagittal images), thus lacking validation and generalizability. Beyond the midbrain, there are other brain structures involved in PSP which can be readily measured on MR images. There is evidence from several sonographic [37,38] and MRI studies [39,40] that the third ventricle (3rdV) is typically enlarged in PSP and progressively dilates over time [40,41], while it is usually spared in early-stage PD patients [39]. The 3rdV width normalized by the internal skull diameter (3rdV/ID) has been recently proposed as a simple manual measurement to help in the differential diagnosis between PSP and PD [39]. This biomarker is extremely simple and can be manually measured on routine axial slices at the level of the 3rdV maximal dilatation [39], thus configuring it as a first-level biomarker to be used in clinical settings where complex biomarkers requiring technology and expertise are not available. The 3rdV/ID showed high reproducibility and good diagnostic performances in distinguishing between PSP and PD (area under the receiver operating characteristic curve (AUC) >0.90) in a single-center cohort [39]. Of high importance, these results were validated in a large independent international cohort of early-stage PSP patients and de novo PD patients, demonstrating the generalizability of the results and the usefulness of this biomarker also in the early stages of the diseases [39]. Thus, in de novo PD patients without marked cognitive impairment, the presence of a small third ventricle helps consolidate the clinical diagnosis, making PSP unlikely; on the contrary, the enlargement of this brain structure (3rdV/ID ≥ 5.88 in the original study [39]) may raise the suspicion of an alternative diagnosis, such as PSP or normal pressure hydrocephalus. The simple manual linear measurements of the M/P diameter ratio and 3rdV/ID ratio performed according to their original descriptions are shown in Figure 2.

### 2.2. Simple Manual Non-Linear Measurements

In contrast to the modest number of studies investigating the role of the midbrain and M/P diameter in the differential diagnosis between PSP and PD, a much greater effort has been made to evaluate the classification performance of the M/P area ratio, a biomarker described in 2005 by Oba and colleagues [42] based on the measurement of midbrain and pons areas on the midsagittal T1-weighted images. This planimetric biomarker is more time-consuming than the linear measurements and probably less suitable for an ambulatory setting, but several data confirmed its usefulness in the differential diagnosis between PSP and PD [40,41,43,44,45,46,47,48,49,50,51,52,53,54,55]. The M/P (or P/M) area ratio showed sensitivity and specificity above 85% in most studies, ranging from 60–65% in some reports to 100% in others [32,33,43,44,45,46,47,48,49,50,51,52,53,54,55]. A list of the studies investigating the performances of midbrain diameter, midbrain area, and M/P or P/M area ratio is shown in Table 1.

### 2.3. Combined MR Planimetric Biomarkers

One of the most robust MR biomarkers for the diagnosis of PSP is the Magnetic Resonance Parkinsonism Index (MRPI), developed in 2008 by Quattrone and colleagues [43]. This combined (including linear and non-linear measurements) planimetric biomarker included in the calculation of the M/P area ratio proposed by Oba et al. [42], together with the linear measurement of the SCP width (measured in its posterior portion) normalized by the middle cerebellar peduncle (MCP) (MRPI = [pons area/midbrain area] · [middle cerebellar peduncle width/superior cerebellar peduncle width]) [43]. All these measurements can be performed on volumetric T1-weighted MR images, with the midbrain area and pons area measured on a midsagittal slice, the MCP width measured on a parasagittal slice, and the SCP width measured on an oblique coronal slice tangent to the floor of the fourth ventricle. Most studies so far calculated the MRPI on volumetric T1-weighted MR images with a voxel size ranging from 0.5 to 1.2 mm^3^ and no slice gap. The inclusion of the SCP measurement led to higher accuracy of MRPI than the M/P area ratio in most studies (Table 2), and the MRPI appears to be less affected by aging compared with the MP area ratio [56,57]. The rationale for including the SCP width came from neuroimaging [43,58] and pathological evidence [59] of selective SCP involvement in PSP patients, with progressive atrophy of this structure over time [58]. Differently from the simple measures discussed above, however, the manual measurement of the SCP width is time-consuming and difficult to perform for non-expert raters [33,53], introducing some variability in the measurements across centers and highlighting the need for automated calculation procedures. For this reason, an automated version of MRPI [60] was developed a few years ago, providing a fast and observer-independent approach to reduce the variability of manual measurements that need expertise for image reconstruction and slice selection. In an Italian multicenter study [60], the automated MRPI showed a high correlation (r = 0.91, *p* < 0.001) with the manual measurements performed by an expert rater, with no difference between 3 T and 1.5 T scanners, and showed high diagnostic performance in distinguishing between PSP and PD. The automated MRPI was recently validated in a multicenter international study [61], including 173 patients affected by PSP, 283 patients with PD, 52 patients with MSA, and 148 healthy controls from several research centers in different countries, and showed excellent diagnostic accuracy (AUC 0.95, confidence intervals [CI]: 0.93–0.97) in distinguishing PSP from non-PSP participants (PD, MSA, and controls). In the last few years, the role of MRPI in the differential diagnosis between PSP and PD was confirmed by many studies [43,44,45,46,47,48,49,50,52,53,54,55,61,62,63] and strengthened by two metanalyses [64,65] including studies from different research groups for a total of nearly 500 PSP patients and more than 1000 PD patients. These metanalyses revealed excellent pooled sensitivity and specificity of MRPI (0.96 and 0.98, respectively, in one study [64], and 0.98 and 0.99, respectively, in another one [65]) and confirmed its superiority compared to the M/P area ratio in differentiating between these two diseases [65], thus recommending the use MRPI in PSP diagnosis.

In most studies, the MRPI and M/P area ratio showed high diagnostic accuracy in differentiating PSP-RS from PD (Table 2). However, some evidence showed suboptimal (<80%) sensitivity of these biomarkers in distinguishing PSP-P and other PSP variants from PD [45,55,62], probably due to the lower degree of brain atrophy in milder PSP subtypes (Table 3). To overcome this limitation, a new version of the MRPI (termed MRPI 2.0) has been recently developed [62]. In addition to the brainstem structures measured by MRPI (midbrain, pons, middle and superior cerebellar peduncles), the MRPI 2.0 also includes in the calculation the measurement of the 3 V width, a brain structure that is severely enlarged in PSP but spared in PD patients [62]. The MRPI and MRPI 2.0 measurements performed according to their original descriptions are shown in Figure 3, and a list of the studies investigating the performances of these two biomarkers is shown in Table 2 (PSP-RS versus PD) and Table 3 (PSP-P versus PD).

The MRPI 2.0 showed higher sensitivity and accuracy than MRPI in distinguishing PSP-P patients from PD (MRPI 2.0, sensitivity: 1.0, accuracy: 0.97; MRPI sensitivity 0.73, accuracy 0.88 in the original study [62]), and the advantage of using MRPI 2.0 was even higher in PSP-P patients with mild ocular slowness (before the appearance of VSGP), which are in an early stage of the disease (MRPI 2.0 accuracy: 0.96; MRPI accuracy: 0.82) [62]. These data suggest that the MRPI can be used in distinguishing PSP-RS from PD, while the MRPI 2.0 is more suitable for differentiating PSP-P from PD. Very recently, an automated version of MRPI 2.0 has been developed [68] to standardize measurements across centers, and the automated MRPI 2.0 was tested with excellent results in two independent large cohorts of PSP-P and PD patients, validating this new biomarker (AUC of 0.93, CI: 0.89–0.98 in a single-center Italian cohort and AUC of 0.92, CI: 0.87.0.96 in an independent multicenter international cohort) [68]. In this study [68], the automated algorithm was strongly correlated (r = 0.91, *p* < 0.001) with the manual measurements performed by an expert rater and showed perfect reproducibility (inter-rater correlation coefficient, ICC = 1) when the automated process was repeated twice starting from the raw 3D T1-weighted images. The high reproducibility guaranteed by the automated software improves the reliability of this biomarker and makes it suitable for longitudinal studies aiming at evaluating the disease progression of PSP through repeated measurements over time. To promote the use of these biomarkers in different countries, a free online platform with a user-friendly interface to calculate the automated MRPI and MRPI 2.0 is currently available at https://mrpi.unicz.it (accessed on 13 July 2022) upon registration, making these automated planimetric biomarkers widely available worldwide. The optimal cut-off values of automated MRPI in distinguishing PSP-RS from PD and of MRPI 2.0 in distinguishing PSP-P from PD derived from the largest international cohorts are 13.88 for MRPI (89% sensitivity, 95% specificity, 94% accuracy) and 2.70 for MRPI 2.0 (86% sensitivity, 92% specificity, 90% accuracy). The cut-off of 13.88 for automated MRPI (PSP-RS vs. PD) was originally identified in a study by Quattrone et al. [62] and validated in a recent international study by Nigro et al. [61]; the cut-off of 2.70 for automated MRPI 2.0 (PSP-P vs. PD) was the optimal cut-off calculated in an international cohort of 56 PSP-P, and 166 PD enrolled in a recent study [68]. The MRPI cut-off value derived from the ROC analysis corresponds well to the 95th percentile MRPI value obtained in a very large cohort of nearly 1000 control subjects in an independent study [57], demonstrating that MRPI values higher than 13.88 reflects a degree of the midbrain and SCP atrophy usually not found in the normal population. In addition to the MRPI and MRPI 2.0 optimal cut-offs derived by the ROC analyses, Figure 4 shows cut-off values corresponding to different sensitivity and specificity thresholds and the probability of having PSP rather than PD for each MRPI and MRPI 2.0 value, which allows adopting a tailored approach in the use of these biomarkers for clinical trials and research studies in PSP and PD patients.

### 2.4. Role of Planimetric Biomarkers in Distinguishing PSP from Other Atypical Parkinsonisms

Most neuroimaging studies investigated the diagnostic performances of MR planimetric measures in differentiating PSP-RS and PSP-P from PD, which is the main focus of this review. Some studies, however, also included patients with the parkinsonian variant of MSA (MSA-P). A recent meta-analysis [69] summarized the current evidence regarding the accuracy of M/P and MRPI in distinguishing PSP (mainly PSP-RS) from the parkinsonian variant of MSA (MSA-P). These planimetric brainstem measurements yielded good diagnostic accuracy for the discrimination of PSP from MSA, with the M/P area ratio showing higher sensitivity and the MRPI showing higher specificity (MRPI: 79.2% pooled sensitivity and 91.2% pooled specificity; M/P: 84.1% pooled sensitivity and 89.2% pooled specificity). Only one study [70] investigated the diagnostic performance of planimetric biomarkers in distinguishing the PSP-P subtype from MSA-P, showing some overlap between groups. Further studies are needed to better elucidate the role of MRPI and MRPI 2.0 in distinguishing PSP-P from MSA-P.

Very few data also exist on the differential diagnosis between PSP and cortico-basal degeneration (CBS). The midbrain is generally more atrophic in PSP-RS than in CBS at a group level [71], while no differences were found between PSP-P and CBS in one study [70]. This overlap can be partially explained because the clinical diagnosis of CBS is associated with a bucket of different underlying pathologies, including CBD, Alzheimer’s disease, TDP-43 pathology but also PSP pathology in a significant number of cases [67,72], thus making the pathological confirmation absolutely needed in this context. Very recently, a large pathological study [67] demonstrated that MRPI yielded good performances (AUC of 0.83) in differentiating between pathologically confirmed PSP-RS and CBD (see below, paragraph 2.6). Further pathological studies are warranted to investigate the role of planimetric biomarkers in distinguishing between PSP-P and CBD.

### 2.5. Role of Planimetric Biomarkers in the Early Stage of the Diseases

The clinical differential diagnosis between PD and PSP is mostly guided by the presence of atypical signs suggestive of PSP, such as vertical oculomotor dysfunction. According to the MDS criteria for PSP [11], the diagnosis of probable or possible PSP (in most subtypes) requires the presence of VSGP or ocular slowness, thus not allowing a clinical diagnosis of PSP before the appearance of oculomotor dysfunction. For this reason, a common clinical scenario involves patients initially diagnosed as PD with the diagnosis refined over time; in a large clinicopathological study [4] carried out in a third-level referral center in the UK, more than one-third (36%) of patients initially classified as PD changed diagnosis at follow-up, due to the appearance of atypical clinical signs during the course of the disease. In an effort to improve the sensitivity of clinical PSP diagnosis in the early stages, a new diagnostic category termed “suggestive of PSP” has been included in the MDS criteria [11] for patients with some clinical features of PSP but lacking overt ocular saccadic impairment. This new diagnostic category, however, showed low specificity for PSP pathology [66,73], including several patients with other neurodegenerative conditions such as PD, multiple system atrophy, cortico-basal degeneration, and dementia.

In these patients with parkinsonism and mild atypical features, often not severe or early enough to exclude a diagnosis of PD, imaging biomarkers may play a crucial role in improving the differential diagnosis between PSP and PD. Evidence in this regard is limited since the evaluation of imaging biomarkers in these patients requires longitudinal studies with clinical follow-up or post-mortem confirmation due to the lack of gold standard examinations to confirm the diagnosis in vivo. However, a few longitudinal studies did investigate MR planimetric biomarkers in patients with clinically unclassifiable parkinsonism (CUP) [74,75] and in patients clinically indistinguishable from PD who later developed a PSP phenotype [76], demonstrating that imaging findings could precede the appearance of oculomotor dysfunction and predict a PSP diagnosis before patients fulfilled the clinical diagnostic criteria. One study by Morelli et al. [74] in 2011 demonstrated in a group of 45 CUP patients that MRPI was able to distinguish patients who later developed a PSP phenotype from those who did not with a 92.8% accuracy. A recent study by Heim et al. [75] investigated 84 patients with parkinsonism who did not fulfill clinical diagnostic criteria for PD, MSA, or PSP at the time of the first evaluation and were followed up for at least two years. MR planimetric biomarkers were able to accurately distinguish at baseline patients who later developed PSP signs from those who later fulfilled MSA or PD criteria, with MRPI 2.0 showing higher classification performances (AUC = 0.98, CI: 0.96–100) than MRPI (AUC of 0.91, CI: 0.82–0.99) [75]. Both biomarkers showed 100% sensitivity for patients who later fulfilled PSP-RS criteria, but MRPI 2.0 yielded significantly higher sensitivity than MRPI (100% vs. 57.1%) for patients who later fulfilled PSP-P criteria [75]. In another recent study by Quattrone et al. [76], 10 patients out of a cohort of 110 patients with an initial diagnosis of PD developed vertical gaze dysfunction during a 4-year follow-up, refining the diagnosis to PSP-P, and MRPI 2.0 was able to identify all these patients at baseline, distinguishing them from patients who maintained the PD diagnosis at follow-up. In accordance with the study by Heim et al. [75], also in this study [76], the MRPI 2.0 was more powerful than MRPI and M/P area ratio in predicting PSP-P diagnosis. These data, if confirmed in further longitudinal studies, demonstrate that the MRPI 2.0 is an early diagnostic MR biomarker for PSP, able to predict the future appearance of a PSP phenotype in patients clinically indistinguishable from PD. Overall, these results may provide considerable help in the differential diagnosis between these two diseases and in the enrollment of early-stage PSP patients in trials with new disease-modifying therapies.

### 2.6. MR Planimetric Biomarkers and PSP Pathology

The MRPI and MRPI 2.0 are considered reliable biomarkers to support the clinical differential diagnosis between PSP and PD. It is still not known, however, if these imaging biomarkers reflect the PSP clinical syndrome or track the underlying pathology regardless of the clinical presentation. There is plenty of evidence in PSP for a mismatch between neuropathology and clinical presentation, which contributes to the suboptimal clinical diagnostic accuracy of parkinsonian syndromes. There are several reports of patients with a clinical PSP phenotype associated with non-PSP pathologies such as cortico-basal degeneration pathology (CBD) [67,72,73,77,78,79], globular glial tauopathies [73,78], TDP-43 pathology [79,80], and alpha-synucleinopathies [77,78,81,82]. On the other hand, patients with post-mortem PSP diagnosis can at times present in the early stages with parkinsonian syndromes resembling PD or MSA [73,82,83], cortico-basal syndrome [67,72], behavioral variant of fronto-temporal dementia [67,72,84,85,86], primary progressive aphasia [67,87], and rarely amnestic syndrome [67]. In this context of clinico-pathological heterogeneity, a biomarker associated with the underlying pathology would be of extreme value to guide the clinical diagnosis and also the selection of patients for clinical trials with new therapies directed to molecular targets.

In 2013, a small study [88] evaluated the midbrain area in a few patients with pathologically proven PSP or CBD. There was a large clinical overlap between CBD and PSP since both diseases can present with cortico-basal syndrome or a PSP phenotype. The authors evaluated different phenotype-pathology combinations and found that the midbrain area was reduced in patients with clinical PSP phenotype, regardless of the underlying disease (PSP or CBD) documented at post-mortem examination, thus suggesting that the midbrain atrophy may not be a specific biomarker of PSP pathology. On the other hand, two very recent larger studies [67,78] demonstrated the usefulness of MRPI in patients with neurodegenerative disorders in predicting the presence of PSP pathology. In one study [78], the MRPI showed abnormal values in 24 out of 29 definite PSP patients, while it was normal in 12 out of 14 patients with a clinical PSP phenotype due to other non-PSP underlying pathologies (alpha-synucleinopathies, CBD, multisystem tauopathy with globular glial inclusions and mitochondrial disorders). In this study [78], the PSP clinical diagnosis yielded a positive predictive value (PPV) of 67.4% for PSP pathology, while the evaluation of MRPI together with the clinical diagnosis raised the PPV for PSP pathology from 67.4% to 92.3%. When compared with other MR planimetric biomarkers, the MRPI was more powerful than the M/P area ratio and midbrain area alone in predicting the underlying pathology [78]. Moreover, another very recent report showed that a few atypical MSA patients presenting with an overt PSP clinical phenotype had normal MRPI and M/P area ratio values [82]. No study so far has investigated the MRPI in patients with a PSP phenotype due to vascular pathology. Taken together, these data demonstrated that a patient with a clinical PSP phenotype and a high MRPI value is extremely likely to have PSP pathology, while a low MRPI value in patients with clinical signs of PSP points toward an alternative underlying pathological condition mimicking this disease. Another large study [67] demonstrated that MRPI accurately distinguished (AUC of 0.90, CI: 0.86–0.95) pathologically proven PSP patients (*n* = 68) from other neurodegenerative disorders (CBD, fronto-temporal lobe degeneration-TDP, Pick’s disease, Alzheimer’s disease, alpha-synucleinopathies, and motor neuron disease-TDP). Of note, MRPI also showed reasonably high accuracy (AUC of 0.83, CI: 0.76–0.90) in differentiating between two 4R-tauopathies (PSP-RS and CBD) [67]. In this study [67], patients were classified according to the pathological diagnosis regardless of the clinical phenotype, and a large percentage (37%) of pathologically proven PSP patients did not present with a clinical PSP phenotype at the time of MRI. Taken together, these recent findings demonstrate that high MRPI values were strongly associated with PSP pathology, regardless of clinical presentation.

## 3. Conclusions

A plethora of studies demonstrated the usefulness of planimetric measures for the differentiation of PSP from PD. Simple and accurate planimetric measurements such as the midbrain diameter or the 3rdV/ID may play a role in ambulatory clinical settings or primary care as first-level tools to support the clinical differential diagnosis between PSP and PD in everyday clinical practice. On the other hand, automated combined planimetric measurements may be more appropriate for distinguishing different PSP phenotypes from PD and also for patients at the early stage of the disease before they fulfill international criteria for PSP diagnosis.

MRPI and MRPI 2.0 are reliable and validated automated biomarkers for the differential diagnosis between PSP and PD. According to the recent classification of levels of evidence for neuroimaging biomarkers in PSP [21], these biomarkers fulfill the “level 2” of the classification, meaning that they can support the clinical diagnosis at the individual level. There is also recent evidence, however, that these biomarkers could go beyond clinical diagnosis. First, MRPI and MRPI 2.0 showed utility in predicting the appearance of a PSP phenotype in patients with parkinsonism before they develop oculomotor dysfunction and meet the clinical diagnostic criteria for possible or probable PSP, thus fulfilling level 3 of the classification [21] (“supportive of early clinical diagnosis”). Second, very recent studies [67,78,82] showed that MRPI had a strong relationship with the underlying PSP pathology regardless of the clinical phenotype. This latter evidence pushes the MRPI forward to level 4 of the classification [21], including neuroimaging biomarkers “Supportive of pathological diagnosis”. Taken together, these data demonstrate that MRPI and MRPI 2.0 may have a crucial role in improving the differential diagnosis between PSP and PD in vivo and provide a strong impetus to use these automated biomarkers in clinical practice. The free and easy-to-use online platform currently available at https://mrpi.unicz.it (accessed on 13 July 2022) could increase the availability of these automated planimetric biomarkers worldwide and may allow using them in multicenter studies and in the selection of early-stage patients for clinical trials with potential new disease-modifying therapies.

Future research is needed to explore the high potential of MR planimetry in combination with other MR techniques (diffusion tensor imaging, neuromelanin-sensitive sequences, iron quantification), with positron emission tomography tau-binding tracers or with fluid biomarkers to further improve the diagnosis of Parkinson’s disease and other parkinsonian syndromes.

## Figures and Tables

**Figure 1 brainsci-12-00949-f001:**
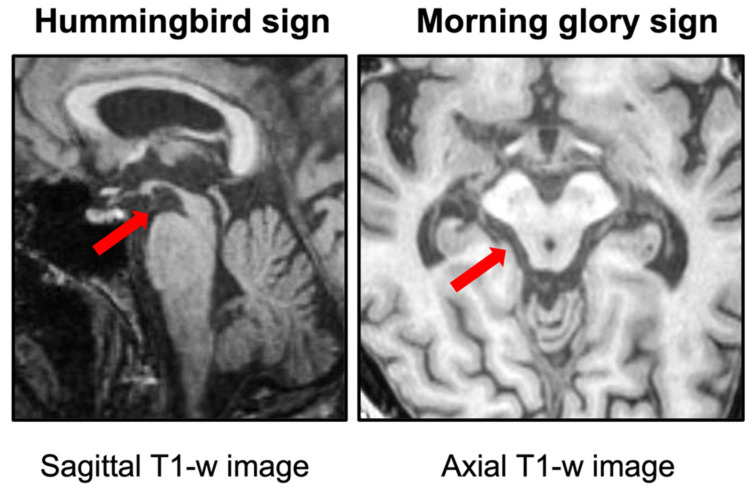
On the left, a T1-weighted midsagittal MR image showing the hummingbird sign with atrophy of the dorsal midbrain (red arrow) and relative sparing of the pons in a patient with progressive supranuclear palsy. On the right, a T1-weighted axial MR image showing the Morning Glory sign with concavity of the lateral margin of the midbrain tegmentum (red arrow) in a patient with progressive supranuclear palsy.

**Figure 2 brainsci-12-00949-f002:**
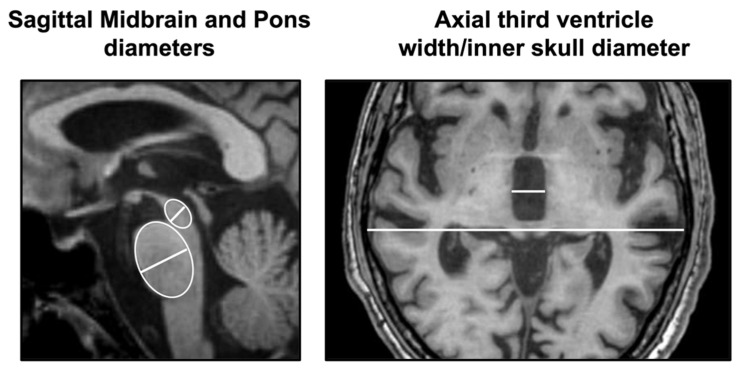
On the left, a midsagittal T1-weighted MR image showing midbrain and pons diameters in a patient with progressive supranuclear palsy. Elliptical regions of interest were placed at their best fit over the midbrain and the pons, and the maximal measurement perpendicular to the major axis of each ellipse was taken (white lines). The midbrain diameter was normalized dividing it by the pons diameter. On the right, a subcallosal axial T1-weighted MR image showing the measurement of the third ventricle width and the internal skull diameter in a patient with progressive supranuclear palsy. Measurements were performed at the level of the third ventricle’s maximum dilatation as the largest left-to-right width between the lateral borders of the ventricle in its central portion. The maximum internal skull diameter (ID) was also measured on the same axial slice. The third ventricle width was normalized dividing it by the ID, and the ratio value was multiplied by 100.

**Figure 3 brainsci-12-00949-f003:**
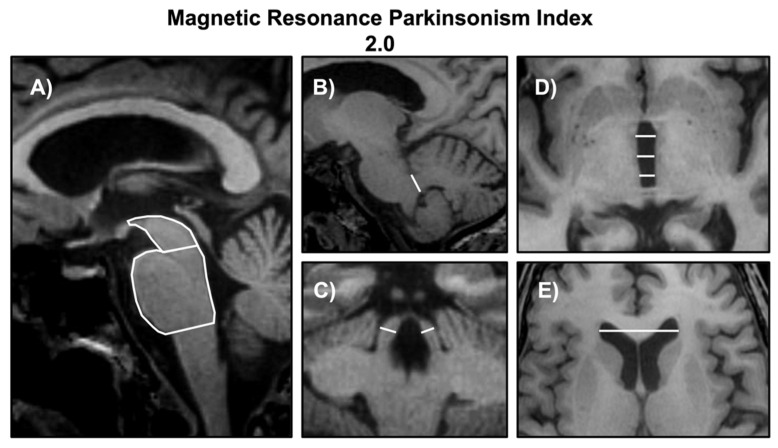
Manual measurements of the midbrain and pons area (**A**), the middle cerebellar peduncles (MCP) width (**B**), the superior cerebellar peduncles (SCP) width (**C**), the third ventricle width (**D**), and the frontal horns width (**E**) on T1-weighted magnetic resonance images in a patient with Parkinson’s disease. The measurements of midbrain and pons area (**A**) were performed on a midsagittal slice. The measurement of the MCP width (**B**) was performed on 3 consecutive parasagittal slices between the pons and the cerebellum for each side. The measurement of the SCP width (**C**) was performed bilaterally on 2 consecutive coronal slices where the peduncles and the inferior colliculi were separated. The measurement of the third ventricle width (**D**) was performed on a bi-commissural axial slice at the level of the anterior and posterior commissures; three measurements were performed at the level of the anterior, middle, and posterior sections of the third ventricle on the same axial slice. The measurement of the width of the frontal horns of lateral ventricles was performed on an axial slice at the level of their maximal dilatation.

**Figure 4 brainsci-12-00949-f004:**
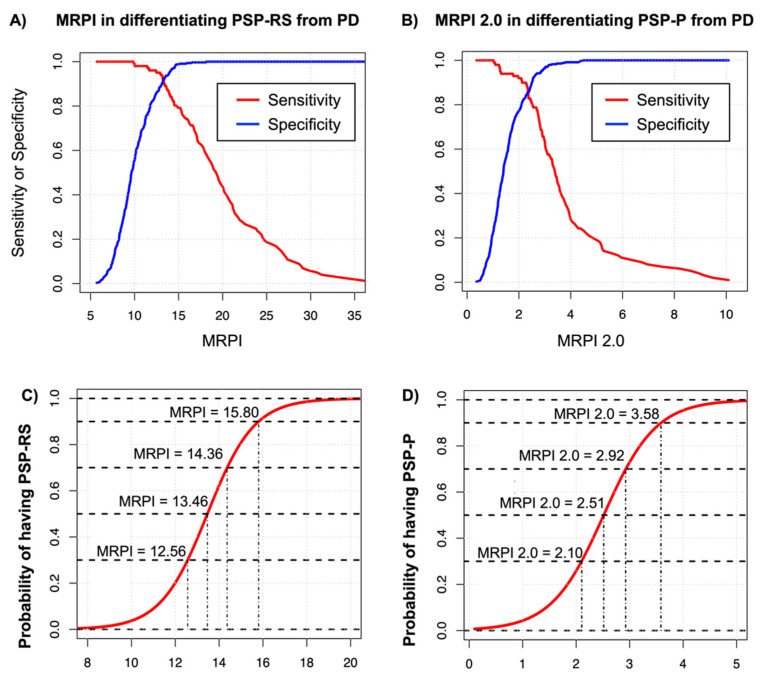
Abbreviations: PSP-RS—progressive supranuclear palsy-Richardson’s syndrome; PSP-P —PSP-Parkinsonism; PD—Parkinson’s disease; MRPI—Magnetic Resonance Parkinsonism Index. At the top are shown the automated MRPI (**A**) and automated MRPI 2.0 (**B**) cut-off values corresponding to different sensitivity and specificity thresholds. At the bottom, there is the probability of having PSP-RS rather than PD for each automated MRPI value (**C**) and the probability of having PSP-P rather than PD for each automated MRPI 2.0 value (**D**), obtained using logistic regression models balancing the number of PD and PSP patients. Data for the PSP-RS versus PD comparisons were obtained in the international cohort enrolled in the study by Nigro et al., 2020 [61]; data for the PSP-P versus PD comparisons were obtained in the international cohort enrolled in the study by Quattrone et al., 2022 [68].

**Table 1 brainsci-12-00949-t001:** Studies that assessed the diagnostic performance of brainstem simple manual MR planimetric measurements in distinguishing between progressive supranuclear palsy-Richardson’s syndrome and Parkinson’s disease.

Study	Patients	MR Biomarker	Cut-Off Value	Sensitivity	Specificity	AUC
Midbrain diameter						
Kim et al., 2015 [32]	29 PSP-RS vs. 82 PD	Midbrain diameter (axial)	≤0.35	50.0	85.2	0.73
Warmuth-Metz et al., 2000 [30]	16 PSP-RS vs. 20 PD	Midbrain diameter (axial)	/	**100**	**100**	/
Owens et al., 2016 [31]	25 PSP-RS vs. 25 PD	Midbrain diameter (sagittal)	/	76	100	/
Owens et al., 2016 [31]	25 PSP-RS vs. 25 PD	M/P diameter ratio (sagittal)	/	**80**	**100**	/
Midbrain area						
Zanigni et al., 2016 [47]	23 PSP-RS vs. 42 PD	Midbrain area	≤102.5	**96.0**	**98.0**	0.99
Moller et al., 2017 [33]	106 PSP-RS vs. 204 PD	Midbrain area	≤124	**83.0**	**83.8**	0.90
Ahn et al., 2019 [51]	27 PSP-RS vs. 27 PD	Midbrain area	≤96	77.8	100	/
M/P or P/M area ratio						
Quattrone et al., 2008 [42]	33 PSP-RS vs. 108 PD	P/M area ratio	≥4.88	**90.9**	**93.5**	/
Hussl et al., 2010 [44]	22 PSP-RS vs. 75 PD	M/P area ratio	≤0.18	63.6	94.7	/
Longoni et al., 2011 [45]	10 PSP-RS vs. 25 PD	P/M area ratio	≥6.01	**90.0**	**96.0**	/
Morelli et al., 2011 [46]	42 PSP-RS vs. 170 PD	M/P area ratio	≤0.20	**92.9**	**85.3**	/
Kim et al., 2015 [32]	29 PSP-RS vs. 82 PD	M/P area ratio	≤0.18	61.5	72.1	0.71
Zanigni et al., 2016 [47]	23 PSP-RS vs. 42 PD	P/M area ratio	≥4.79	**96.0**	**90.0**	0.97
Sankhla et al., 2016 [48]	26 PSP-RS vs. 13 PD	M/P area ratio	<0.21	**100**	**92.9**	/
Nigro et al., 2017 [49]	15 PSP-RS vs. 179 PD ^a^	M/P area ratio	≤0.15	**100**	**100**	/
Nigro et al., 2017 [49]	20 PSP-RS vs. 179 PD ^b^	M/P area ratio	≤0.19	**85.0**	**93.8**	/
Nizamani et al., 2017 [50]	34 PSP-RS vs. 34 PD	P/M area ratio	≥4.20	**85.7**	**91.7**	/
Moller et al., 2017 [33]	106 PSP-RS vs. 204 PD	M/P area ratio	≤0.21	76.4	80.4	0.84
Ahn et al., 2019 [51]	27 PSP-RS vs. 27 PD	P/M area ratio	≥5.13	**88.9**	**100**	/
Nakahara et al., 2019 [52]	26 PSP-RS vs. 93 PD	P/M area ratio	≥4.22	100	65.6	/
Oktay et al., 2020 [53]	14 PSP-RS vs. 43 PD	P/M area ratio	≥4.51	78.0	70.0	/
Sjöström et al., 2020 [54]	29 PSP-RS vs. 104 PD	M/P area ratio	/	75.9	83.6	0.81
Picillo et al., 2020 [55]	38 PSP-RS vs. 35 PD	P/M area ratio	≥4.97	**94.4**	**88.6**	0.96

Note to Table 1: Studies showing both sensitivity and specificity above 80% are highlighted in bold. Studies aiming to differentiate PSP from MSA or non-PSP patients (including combinations of PD, MSA, and controls) and studies with sample sizes smaller than 10 patients were not included in the table. Abbreviations: PSP-RS—progressive supranuclear palsy Richardson’s syndrome; PD—Parkinson’s disease; AUC—area under the receiver operating characteristic curve; M/P—midbrain/pons; P/M—pons/midbrain. ^a^ Probable PSP patients according to NINDS-SPSP criteria [10]. ^b^ Possible PSP patients according to NINDS-SPSP criteria [10].

**Table 2 brainsci-12-00949-t002:** Studies that assessed the diagnostic performance of combined MR planimetric biomarkers in distinguishing between progressive supranuclear palsy-Richardson’s syndrome and Parkinson’s disease.

Study	Patients	MR Biomarker	Cut-Off Value	Sensitivity	Specificity	AUC
MRPI						
Quattrone et al., 2008 [43]	33 PSP-RS vs. 108 PD	Manual MRPI	≥13.55	**100**	**100**	/
Hussl et al., 2010 [44]	22 PSP-RS vs. 75 PD	Manual MRPI	≥14.38	81.8	76.0	/
Longoni et al., 2011 [45]	10 PSP-RS vs. 25 PD	Manual MRPI	≥13.57	**100**	**92**	/
Morelli et al., 2011 [46]	42 PSP-RS vs. 170 PD	Manual MRPI	≥13.60	**100**	**99.4**	/
Kim et al., 2015 [32]	29 PSP-RS vs. 82 PD	Manual MRPI	>8.92	92.3	39.7	0.66
Zanigni et al., 2016 [47]	23 PSP-RS vs. 42 PD	Manual MRPI	≥10.67	**87.0**	**93.0**	0.95
Nigro et al., 2017 (3 T) [60]	38 PSP-RS vs. 156 PD	Manual MRPI	≥13.37	**100**	**100**	/
Nigro et al., 2017 (1.5 T) [60]	50 PSP-RS vs. 78 PD	Manual MRPI	≥13.43	**90.9**	**100**	/
Sankhla et al., 2016 [48]	26 PSP-RS vs. 13 PD	Manual MRPI	>12.4	**100**	**100**	/
Moller et al., 2017 [33]	106 PSP-RS vs. 204 PD	Manual MRPI	>8.98	64.2	64.2	0.75
Nigro et al., 2017 [49]	15 PSP-RS vs. 179 PD ^a^	Manual MRPI	≥15.64	**100**	**100**	/
Nigro et al., 2017 [49]	20 PSP-RS vs. 179 PD ^b^	Manual MRPI	≥13.38	**100**	**98.9**	/
Nizamani et al., 2017 [50]	34 PSP-RS vs. 34 PD	Manual MRPI	≥13.50	**100**	**100**	/
Nakahara et al., 2019 [52]	26 PSP-RS vs. 93 PD	Manual MRPI	≥11.30	73.1	68.8	/
Oktay et al., 2020 [53]	14 PSP-RS vs. 43 PD	Manual MRPI	≥13.63	78.0	82.0	0.87
Sjöström et al., 2020 [54]	29 PSP-RS vs. 104 PD	Manual MRPI	/	65.5	84.3	0.77
Picillo et al., 2020 [55]	38 PSP-RS vs. 35 PD	Manual MRPI	≥13.89	**86.8**	**91.4**	0.93
Kim et al., 2021 (metanalysis) [64]	484 PSP vs. 1243 PD	Manual MRPI ^c^	/	**96.0**	**98.0**	0.99
Nigro et al., 2017 (3 T) [60]	38 PSP-RS vs. 156 PD	Automated MRPI	≥13.42	**97.3**	**97.4**	/
Nigro et al., 2017 (1.5 T) [60]	50 PSP-RS vs. 78 PD	Automated MRPI	≥13.42	**93.2**	**97.3**	/
Nigro et al., 2017 [49]	15 PSP-RS vs. 179 PD ^a^	Automated MRPI	≥15.22	**100**	**98.9**	/
Nigro et al., 2017 [49]	20 PSP-RS vs. 179 PD ^b^	Automated MRPI	≥13.46	**100**	**98.3**	/
Quattrone et al., 2018 [62]	46 PSP-RS vs. 53 PD	Automated MRPI	≥13.88	**100**	**100**	/
MRPI 2.0						
Picillo et al., 2020 [55]	38 PSP-RS vs. 35 PD	Manual MRPI 2.0	≥3.18	**89.5**	**85.7**	0.92
Sjöström et al., 2020 [54]	29 PSP-RS vs. 104 PD	Manual MRPI 2.0	/	72.4	79.3	0.81
Quattrone et al., 2018 [62]	46 PSP-RS vs. 53 PD	Semi-automated MRPI 2.0	≥2.50	**100**	**100**	/

Note to Table 2: Studies showing both sensitivity and specificity above 80% are highlighted in bold. Studies aiming to differentiate PSP from MSA or non-PSP patients (including combinations of PD, MSA, and controls) and studies with sample sizes smaller than 10 patients were not included in the table. Abbreviations: PSP-RS—progressive supranuclear palsy Richardson’s syndrome; PD—Parkinson’s disease; AUC—area under the receiver operating characteristic curve; MRPI—Magnetic Resonance Parkinsonism Index. ^a^ Probable PSP patients according to NINDS-SPSP criteria [10]. ^b^ Possible PSP patients according to NINDS-SPSP criteria [10]. ^c^ The MRPI was measured manually in 12 out of the 14 studies included in the meta-analysis [64].

**Table 3 brainsci-12-00949-t003:** Studies that assessed the diagnostic performance of MR planimetric measurements in distinguishing between progressive supranuclear palsy-Parkinsonism and Parkinson’s disease.

Study	Patients	MR Biomarker	Cut-Off Value	Sensitivity	Specificity	AUC
M/P or P/M area ratio						
Longoni et al., 2011 [45]	10 PSP-P vs. 25 PD *	P/M area ratio	≥6.02	60.0	96.0	/
Quattrone et al., 2019 [66]	10 PSP-P vs. 100 PD	P/M area ratio	≥4.87	**100**	**96**	/
Picillo et al., 2020 [55]	21 PSP-P vs. 35 PD	P/M area ratio	≥4.72	71.4	77.1	0.78
MRPI						
Longoni et al., 2011 [45]	10 PSP-P vs. 25 PD*	Manual MRPI	≥11.07	70.0	68.0	/
Picillo et al., 2020 [55]	21 PSP-P vs. 35 PD	Manual MRPI	≥11.70	65.0	74.3	0.75
Quattrone et al., 2018 [62]	34 PSP-P vs. 53 PD	Automated MRPI	≥12.38	73.5	98.1	/
Quattrone et al., 2019 [67]	10 PSP-P vs. 100 PD	Automated MRPI	≥12.90	**100**	**100**	/
Quattrone et al., 2022 (cohort 1) [68]	43 PSP-P vs. 177 PD	Automated MRPI	≥12.45	76.7	87.6	0.88
Quattrone et al., 2022 (cohort 2) [68]	56 PSP-P vs. 166 PD	Automated MRPI	≥11.84	83.9	77.1	0.86
MRPI 2.0						
Picillo et al., 2020 [55]	21 PSP-P vs. 35 PD	Manual MRPI 2.0	≥2.58	76.2	65.7	0.75
Quattrone et al., 2018 [62]	34 PSP-P vs. 53 PD	Semi-automated MRPI 2.0	≥2.18	**100**	**94.3**	/
Quattrone et al., 2019 [67]	10 PSP-P vs. 100 PD	Semi-automated MRPI 2.0	≥2.88	**100**	**100**	/
Quattrone et al., 2022 (cohort 1) [68]	43 PSP-P vs. 177 PD	Automated MRPI 2.0	≥2.23	**93.0**	**87.6**	0.93
Quattrone et al., 2022 (cohort 2) [68]	56 PSP-P vs. 166 PD	Automated MRPI 2.0	≥2.70	**85.7**	**92.2**	0.92

Note to Table 3: Studies showing both sensitivity and specificity above 80% are highlighted in bold. Studies aiming to differentiate PSP from MSA or non-PSP patients (including combinations of PD, MSA, and controls) and studies with sample sizes smaller than 10 patients were not included in the table. Abbreviations: PSP-P—progressive supranuclear palsy-parkinsonism; PD—Parkinson’s disease; AUC—area under the receiver operating characteristic curve; M/P—midbrain/pons; P/M—pons/midbrain; MRPI—Magnetic Resonance Parkinsonism Index. * PSP-P classified according to expert guidelines [19].

## Data Availability

Not applicable.

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
