# Peer review of "Magnetic Resonance Planimetry in the Differential Diagnosis between Parkinson’s Disease and Progressive Supranuclear Palsy"

_brainsci, 2022, doi:10.3390/brainsci12070949_

Round 1
Reviewer 1 Report
The aim of this review study was to differentiate PD from PSP using MRI T1 weighted imaging for measuring different structures of the brain.
Level of interest: An article whose findings are important to those with closely related research interests.
The article is well written and structured.
The article is based on MRI T1 images but the T1 protocol is not mentioned. What type of acquisition was used? Volumetric (3D), or T1 in 3 different planes. If a non-volumetric acquisition was used, what is the slice thickness, gap, etc. These parameters can influence the final measurements.
There are too many references (98), even for a review paper, please reduce them to maximum 75.
There are also some auto-citations (at least 6). This is also unacceptable.
Author Response
We thank the reviewer for the positive comments. We have included in the manuscript a sentence about the MRI protocol used to calculate the MRPI in most studies, as follows: “[…] All these measurements can be performed on volumetric T1-weighted MR images, with the midbrain area and pons area measured on a midsagittal slice, the MCP width measured on a parasagittal slice, and the SCP width measured on an oblique coronal slice tangent to the floor of the fourth ventricle. Most studies so far calculated the MRPI on volumetric T1-weighted MR images with a voxel size ranging from 0.5 to 1.2 mm3 and no slice gap.”
We could not find any maximum number of references in the “instruction for authors”, but we made a special effort in reducing the number of references and auto-citations as suggested by the reviewer. However, we also had to include some new references in the revised version of our manuscript to address the comments raised by the other reviewers.
Reviewer 2 Report
The article by Quattrone A et al. titled as “Magnetic Resonance planimetry in the differential diagnosis between Parkinson’s disease and progressive supranuclear palsy” highlights the recent development and the role of imaging techniques in the accurate diagnosis of PSP (progressive supranuclear palsy). The usefulness of MR planimetric quantitative biomarkers to differentiate between PD and PSP has been discussed by the authors. This review article will be helpful in understanding the pathology of PSP and PD as well as its therapies and in making clinical decisions. The manuscript is well written and interesting to read.
Author Response
We thank the reviewer. We really appreciated these positive comments.
Reviewer 3 Report
The issue concerning the differentiation between Parkinson's Disease (PD) and Progressive Supranuclear Palsy (PSP) seems crucial, especially in the context of future therapies, which could be more beneficial in the PSP-Parkinsonism Predominant phenotype. In this review authors elaborate on planimetric differential diagnosis of PD and PSP. The review brings several perspectives concerning this issue, however certain points should be changed:
1. In the introduction authors state:
"For example, PSP-P, which is the most frequent phenotype after 55 PSP-RS, has a clinical phenotype mainly characterized by parkinsonism, that can be 56 asymmetric and levodopa-responsive, strongly resembling PD"
It would be valuable to stress that PSP-P in its early stages may also resemble Multiple System Atrophy - Parkinsonism Predominant
Ref.
[A] The Strengths and Obstacles in the Differential Diagnosis of Progressive Supranuclear Palsy-Parkinsonism Predominant (PSP-P) and Multiple System Atrophy (MSA) Using Magnetic Resonance Imaging (MRI) and Perfusion Single Photon Emission Computed Tomography (SPECT). Diagnostics (Basel). 2022 Feb 2;12(2):385. doi: 10.3390/diagnostics12020385. PMID: 35204476; PMCID: PMC8871165.
[B] [Diagnosis of MSA-P and PSP-P in Early Stage]. Brain Nerve. 2020 Apr;72(4):331-343. Japanese. doi: 10.11477/mf.1416201532. PMID: 32284458.
2. In paragraph 2.4. authors state:
"The MRPI and MRPI 2.0 are considered reliable biomarkers to support the clinical 349 differential diagnosis between PSP and other parkinsonian syndromes."
A recent paper [C] stated that MRPI 2.0: "Based on the evaluation of 74 patients, we demonstrate that only the mesencephalon/pons ratio and MRPI show a significant difference between PSP-P and MSA-parkinsonian type (MSA-P). Interestingly, this differential feature was not maintained by MRPI 2.0". The issue concerning the bounded significance of MRPI 2.0 in the differential diagnosis of PSP-P and MSA-P, and the limitations of MRPI, MRPI 2.0 and M/P ratio in the differential diagnosis of tauopathic parkinsonian syndromes should be additionally discussed.
[C] Is MRPI 2.0 More Useful than MRPI and M/P Ratio in Differential Diagnosis of PSP-P with Other Atypical Parkinsonisms? J Clin Med. 2022 May 10;11(10):2701. doi: 10.3390/jcm11102701. PMID: 35628828; PMCID: PMC9147601.
3. Authors state that:
"In 2013, a small study [98] found that the midbrain area was reduced in a few patients 365 with a clinical PSP phenotype, regardless of the underlying disease (PSP or CBD) 366 documented at post-mortem examination, thus suggesting that the midbrain atrophy may 367 not be a specific biomarker of PSP pathology."
It would be valuable to point out not only the various pathologies underlying clinical manifestation, but also the overlapping of clinical manifestations.
4. Another large study [93] demonstrated that MRPI accurately distinguished 382 (AUC of 0.90, CI: 0.86-0.95) pathologically proven PSP patients (n=68) from other 383 neurodegenerative disorders (CBD, fronto-temporal lobe degeneration-TDP, Pick’s 384 disease, Alzheimer’s disease, alpha-synucleinopathies and motor neuron disease-TDP).
It should be stressed that the MRPI was not verified among patients with PSP clinical manifestation based on vascular pathology.
5. Minor point:
Two paragraphs have number "2.3".
Author Response
- In the introduction authors state:
"For example, PSP-P, which is the most frequent phenotype after 55 PSP-RS, has a clinical phenotype mainly characterized by parkinsonism, that can be 56 asymmetric and levodopa-responsive, strongly resembling PD"
It would be valuable to stress that PSP-P in its early stages may also resemble Multiple System Atrophy - Parkinsonism Predominant
Ref.
[A] The Strengths and Obstacles in the Differential Diagnosis of Progressive Supranuclear Palsy-Parkinsonism Predominant (PSP-P) and Multiple System Atrophy (MSA) Using Magnetic Resonance Imaging (MRI) and Perfusion Single Photon Emission Computed Tomography (SPECT). Diagnostics (Basel). 2022 Feb 2;12(2):385. doi: 10.3390/diagnostics12020385. PMID: 35204476; PMCID: PMC8871165.
[B] [Diagnosis of MSA-P and PSP-P in Early Stage]. Brain Nerve. 2020 Apr;72(4):331-343. Japanese. doi: 10.11477/mf.1416201532. PMID: 32284458.
We thank the reviewer for this suggestion. We rephrased the sentence as follows “, PSP-P, which is the most frequent phenotype after PSP-RS, has a clinical phenotype mainly characterized by parkinsonism, making the differential diagnosis with PD and the parkinsonian variant of multiple system atrophy extremely challenging” and we included the suggested reference.
- In paragraph 2.4. authors state:
"The MRPI and MRPI 2.0 are considered reliable biomarkers to support the clinical 349 differential diagnosis between PSP and other parkinsonian syndromes."
A recent paper [C] stated that MRPI 2.0: "Based on the evaluation of 74 patients, we demonstrate that only the mesencephalon/pons ratio and MRPI show a significant difference between PSP-P and MSA-parkinsonian type (MSA-P). Interestingly, this differential feature was not maintained by MRPI 2.0". The issue concerning the bounded significance of MRPI 2.0 in the differential diagnosis of PSP-P and MSA-P, and the limitations of MRPI, MRPI 2.0 and M/P ratio in the differential diagnosis of tauopathic parkinsonian syndromes should be additionally discussed.
[C] Is MRPI 2.0 More Useful than MRPI and M/P Ratio in Differential Diagnosis of PSP-P with Other Atypical Parkinsonisms? J Clin Med. 2022 May 10;11(10):2701. doi: 10.3390/jcm11102701. PMID: 35628828; PMCID: PMC9147601.
We have rephrased the sentence as follows: “The MRPI and MRPI 2.0 are considered reliable biomarkers to support the clinical differential diagnosis between PSP and PD”. In addition, although the main focus of this review was the role of MR planimetry in distinguishing between PSP and PD, we have included in the revised version of our manuscript a paragraph and some comments on the differential diagnosis between PSP and other atypical parkinsonisms (MSA and CBS), as suggested by this reviewer.
- Authors state that:
"In 2013, a small study [98] found that the midbrain area was reduced in a few patients 365 with a clinical PSP phenotype, regardless of the underlying disease (PSP or CBD) 366 documented at post-mortem examination, thus suggesting that the midbrain atrophy may 367 not be a specific biomarker of PSP pathology."
It would be valuable to point out not only the various pathologies underlying clinical manifestation, but also the overlapping of clinical manifestations.
We thank the reviewer for this comment. We have included a sentence to clarify the clinical overlap between CBD and PSP in the revised version of our manuscript, as follows: “In 2013, a small study [88] evaluated the midbrain area in a few patients with pathologically proven PSP or CBD. There was a large clinical overlap between CBD and PSP, since both diseases can present with cortico-basal syndrome or a PSP phenotype The authors evaluated different phenotype-pathology combinations and found that the midbrain area was reduced in patients with clinical PSP phenotype, regardless of the underlying disease (PSP or CBD) documented at post-mortem examination, thus suggesting that the midbrain atrophy may not be a specific biomarker of PSP pathology”.
4. Another large study [93] demonstrated that MRPI accurately distinguished 382 (AUC of 0.90, CI: 0.86-0.95) pathologically proven PSP patients (n=68) from other 383 neurodegenerative disorders (CBD, fronto-temporal lobe degeneration-TDP, Pick’s 384 disease, Alzheimer’s disease, alpha-synucleinopathies and motor neuron disease-TDP). It should be stressed that the MRPI was not verified among patients with PSP clinical manifestation based on vascular pathology.
We included in the paragraph a sentence on this point. “No study so far has investigated the MRPI in patients with a PSP phenotype due to vascular pathology”.
- Minor point:
Two paragraphs have number "2.3".
We apologise for the typo. We corrected it.
Round 2
Reviewer 3 Report
I do not have further comments.